# Visual Tracking with FPN Based on Transformer and Response Map Enhancement

**Anping Deng [1,2], Jinghong Liu [1,*], Qiqi Chen [1,2], Xuan Wang [1] and Yujia Zuo [1]**

1   Changchun Institute of Optics, Fine Mechanics and Physics (CIOMP), Chinese Academy of Sciences, Changchun 130033, China; denganping20@mails.ucas.ac.cn (A.D.); chenqiqi20@mails.ucas.ac.cn (Q.C.); ally637@163.com (X.W.); mzjy0617@126.com (Y.Z.)
2   University of Chinese Academy of Sciences, Beijing 100049, China
*   Correspondence: liu1577@126.com

**Abstract:** Siamese network-based trackers satisfy the balance between performance and efficiency for visual tracking. However, they do not have enough robustness to handle the challenges of target occlusion and similar objects. In order to improve the robustness of the tracking algorithm, this paper proposes visual tracking with FPN based on Transformer and response map enhancement. In this paper, a feature pyramid structure based on Transformer is designed to encode robust target-specific appearance features, as well as the response map enhanced module to improve the tracker's ability to distinguish object and background. Extensive experiments and ablation experiments are conducted on many challenging benchmarks such as UAV123, GOT-10K, LaSOT and OTB100. These results show that the tracking algorithm we proposed in this paper can effectively improve the tracking robustness against the challenges of target occlusion and similar object, and thus improve the precision rate and success rate of the tracking algorithm.

**Keywords:** visual object tracking; transformer; response map enhancement; Siamese network

## 1. Introduction

As an important computer vision task, object tracking is widely applied in the fields of UAV application [1], industrial production and manufacturing [2], automatic driving [3] and so on. Specifically, object tracking refers to the process of estimating the visual target trajectory when only an initial state of the target (in a video frame) is available. However, state-of-the-art visual trackers are still unable to handle serious real-world challenges such as severe variations in target appearance, similar object and occlusion. These challenges can reduce the accuracy and robustness of tracking, which may lead to tracking failures.

The mainstream models can be classified into correlation filter-based trackers and Siamese-based trackers. The main purpose of the correlation filter is to estimate the optimal image filter, which can maximize its response to the region of the target when the search area is entered; however, the accuracy and robustness of the tracking task still needs to be improved. Recently, due to the superior feature extraction ability of deep learning, the accuracy of object tracking has made great progress and Siamese network-based trackers have attracted extensive attention in the field of visual tracking.

Siamese-based networks are composed of two independent feature extraction networks. Given the pairs of target and search regions, these two-stream networks compute the same function to produce a similarity map. Then, the search region with the highest similarity to the target template is found by matching the response map to complete the tracking. In early Siamese-based trackers such as SINT [4], the Siamese network was trained to search for the candidate region most similar to the initial object appearance. The recent SiamFC [5] introduces the cross-correlation layer as a fusion tensor and highly improves the accuracy. However, SiamFC lacks a bounding box regression and is required to carry out a multi-scale test, which makes it less elegant. SiamRPN [6] adds the region proposal

extraction sub-network to the Siamese network. By training both the classification branch and regression branch for visual tracking, SiamRPN avoids the time-consuming step of extracting multi-scale feature maps for the object scale invariance. However, the feature expression ability of SiamRPN is still limited due to the feature extraction ability of the shallow network. SiamRPN++ [7] introduced a deeper network, such as ResNet-50, by improving the sampling method [8], but it also finds it difficult to deal with similar object and target occlusion using only the deep feature. In deep learning, low-level features can better represent spatial information such as color, texture and edge and corner points, which is conducive to target location, the determination of boundary box size and aspect ratio, while high-level features pay more attention to semantic information such as categories and attributes, which is more conducive to foreground and background differentiation. The effective fusion of high-level and low-level features can improve tracking accuracy and robustness.

In view of the above, this paper proposes a target tracking algorithm with a response map enhancement module based on the Transformer feature pyramid (TR-Siam). As shown in Figure 1, the framework mainly consists of four simple sub-networks: feature extraction sub-network, feature fusion sub-network, similarity match sub-network and prediction head. The Transformer-based feature pyramid (TFN) is a key feature of the feature fusion sub-network. TFN can efficiently fuse shallow and deep features to achieve feature enhancement. In addition, the response map enhancement module is introduced into the similarity match sub-network to improve the algorithm's foreground and background recognition abilities. Experiments show that the proposed TR-Siam has better robustness in dealing with complex scenes such as similar object, occlusion and viewpoint change. Our main contributions are:

1.  We propose an object tracking algorithm based on Transformer feature pyramid and add a response map enhancement module. Compared with other trackers, the accuracy and robustness of our tracker are greatly improved when dealing with the challenges of similar object and occlusion.

2.  Feature pyramid based on Transformer proposed in this paper establishes global information interaction, efficiently integrates feature information of low and high layers, and effectively improves tracking performance; the proposed response map enhancement module makes full use of the context information of the response map, which improves the ability of the model to distinguish the object from the background, and effectively improve the ability to deal with the interference of similar targets and other complex environments.

3.  Extensive experiments and ablation experiments are conducted on many challenge benchmarks, these prove our proposed method has competitive performance, not only in accuracy but also in speed. When different backbone networks are selected, the frame rate of the algorithm in this paper can reach 45-130FPS, meeting the real-time requirements.

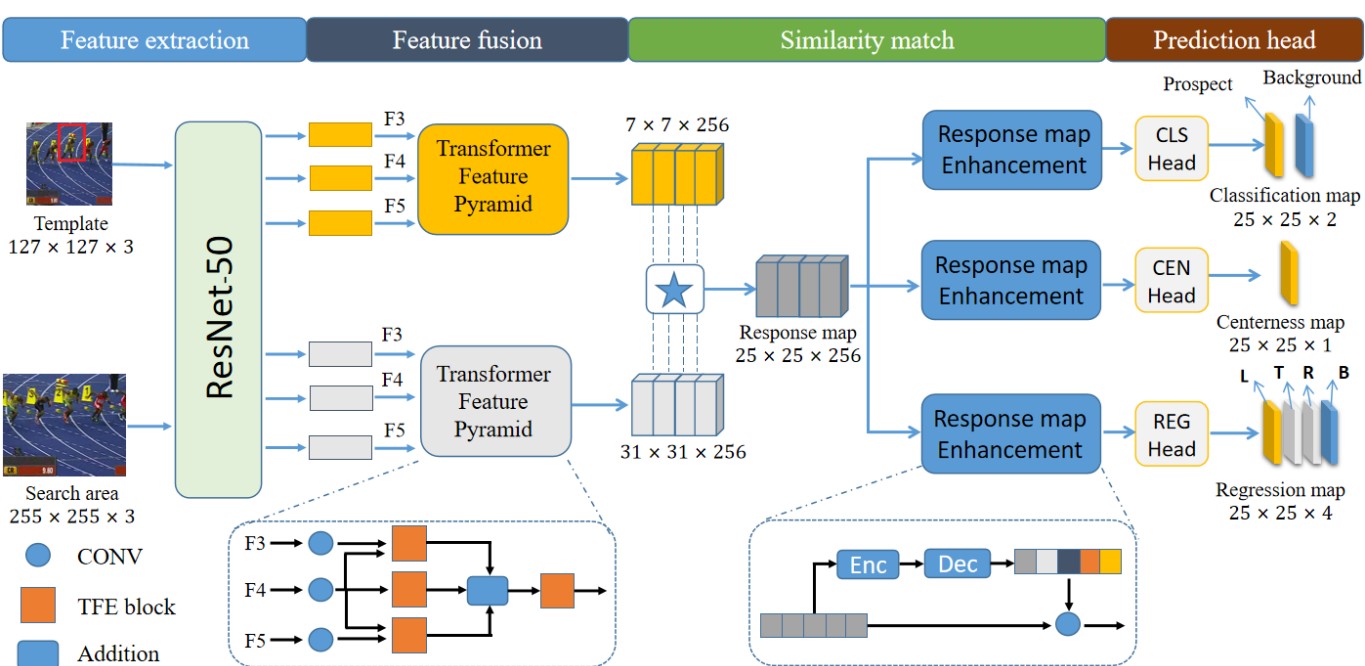

**Figure 1.** This is the schematic diagram of TR-Siam. The ☆ stands for cross-correlation operation. Template contains target of interest. Search area is centered on the template position and three times the size of the template in the detection frame.

## 2. Related Work

In recent years, a large number of excellent trackers have proposed many advanced suggestions. The trackers based on correlation filter use multi-level features, template updating and deeper feature extraction to further improve MOSSE and KCF [9–14], which effectively improve the accuracy and robustness of target tracking.

Based on SiamFC, SiamRPN adopts anchors for region proposal to make use of the deep feature maps and avoid repeated computation, which has great speed performance. Subsequent studies have improved SiamRPN from the aspects of feature extraction, proportion of positive and negative samples, feature enhancement and so on [15,16]. However, the anchor-based trackers are very sensitive to relative hyper-parameters such as size, number and aspect ratio of anchor boxes. Moreover, anchor-based trackers possess limited performance in processing objects with significant appearance change due to their fixed hyper-parameters. Inspired by FCOS, SiamCAR [17] uses an anchor-free framework to reduce the number of hyper-parameters and maintain an excellent performance, both in accuracy and efficiency. In this paper, the anchor-free framework is adopted and SiamCAR is used as our baseline algorithm.

The feature fusion network is a common strategy in visual tracking [18]. To solve the scale variation problem in tracking, the fusion of different deep features are used to exploit complementary spatial and semantic information in SiamFPN [19,20]. SiamHOG fuses the HOG features and deep features of the image [21]. These feature fusion methods, to some extent, improve the tracker's performance when dealing with the target scale variation and small object tracking. However, limited by the receptive field, the above fusion method simply fuses the information of each location with the surroundings, and fails to make effective use of the global context information.

Transformer can overcome receptive field limitations and focus on global information [22]. Inspired by Transformer, HIFT combines Transformer with the feature fusion network [23], which directly fuses different levels of the feature map, but the Transformer encoder and decoder structure is still complex and limited by its adoption of the AlexNet feature extraction network [24]; the tracking accuracy and efficiency still need to be improved. Based on DETR [25], our proposed method designs efficient multi-head attention

layers that are embedded into the feature pyramid through the cross attention between different feature layers to enhance the object features. This then improves its performance when facing challenges such as severe variations in target appearance, similar object and occlusion. In this way, we improve the robustness of the object tracking.

After the backbone models, we can obtain two feature maps of the target template and search region. Siamese-based trackers perform cross-correlation on the feature map of search region with the feature map of target exemplar as the kernel. In this way, we can obtain the response map and send it to the classification and regression branch for visual tracking. However, for the tracker, the template branch and the search branch are always independent, so the template information in the response map always comes from the ground truth of the first frame. So, the tracking may fail when the appearance of an object changes. In this paper, the response map enhancement module is added to enhance the response map by calculating the statistical information of the response map. Specifically, we obtain the context information of the response map to adjust itself. In this way, the useful information between the target template and search area is strengthened and implicitly updates the template to some extent.

## 3. Proposed Method

In this section, we mainly introduce the details of the TFN module and RmE module, and how to combine them with the Siamese network. TR-Siam proposed in this paper consists of four modules: feature extraction sub-network, feature fusion sub-network, similarity match sub-network and prediction head, as shown in Figure 1.

### 3.1. Overall Overview

Our proposed method, TR-Siam, selected ResNet-50 as the backbone for feature extraction. We selected the last three Resblock outputs (Feat3, Feat4, Feat5) of the backbone and sent it to the Transformer-based feature pyramid, which is used to realize the feature fusion of multiple layers. This allowed us to obtain the feature maps of the target exemplar and the search area, which have both location details and semantic information. After that, by performing the cross-correlation on the feature map of the search region with the feature map of the target template as a kernel, we obtained response map R. Then, the response map enhancement module proposed was used to enhance the response map to improve the tracker's discrimination ability to distinguish the target from the background. Finally, it was sent to the prediction head to obtain the tracking results.

### 3.2. Transformer Feature Pyramid

It is generally considered that feature extraction is the key step in visual tracking. During feature extraction, the high-level feature map obtained by convolution and pooling provided a better representation of semantic information. However, for similar objects, the semantic information is similar and difficult to distinguish. In the case of occlusion and small object, the object has small amounts of pixels information, so it is difficult to track well. Feature fusion can effectively combine the advantages of low-level and high-level features: low-level features contain rich spatial information such as color, texture and edge and corner points, which is conducive to locate small and similar objects, but they lack semantic information. On the other hand, high-level features pay more attention to semantic information such as categories and attributes, which is more conducive to foreground and background differentiation. The effective fusion of high-level and low-level features can improve a tracker's ability to deal with challenges such as similar objects, object deformation and scale change. Existing feature fusion methods are mainly feature pyramid, which simply concatenates high-level and low-level feature maps, or uses different level features to accomplish different tasks. However, this method only uses a simple combination of features without global interaction, which is a fundamental limitation.

Based on the limitations of above methods, this paper adopts Transformer-based feature pyramid structure for feature fusion, establishes the connection of global context,

and gives full play to the advantages of feature fusion. We modified the Transformer structure proposed in the literature [25]; specifically, we removed steps of position coding in the traditional Transformer and used zero-padding with its own location information for object tracking. This allowed us to obtain a lighter Transformer-based structure and better accomplish feature fusing. The module is shown in Figure 2.

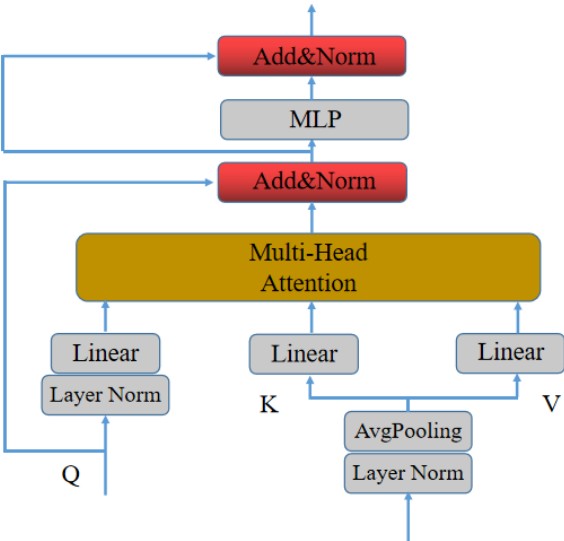

**Figure 2.** Schematic diagram of the proposed TFE block. The module consists of three parts: Multi-Head Attention, Layer Norm and MLP.

The MLP in this figure is the full connected layer, which is used to integrate the feature sequence information. Layer Norm standardizes samples one by one to make the input's distribution stable and avoid the problem of gradient disappearance. The proposed method calculates the input features through the multi-head attention mechanism as follows:

$$
\text{Attention} (Q, \ K, \ V) = \text{Softmax} \left(\frac{Q \cdot K^T}{\sqrt{d_k}}\right) \cdot V
$$
$$
\text{MultiHead}(Q, \ K, \ V) = \text{Concat}(H_1, H_2, \dots, H_N)W^O \tag{1}
$$
$$
H_i = \text{Attention} (QW_i^Q, KW_i^K, VW_i^V)
$$

In this formula, where Q, K, V represent query, key and value; $d_k$ represents the number of channels, i.e., sequence length; N = 8 represents the number of attention heads; $W_i^Q$, $W_i^K$, $W_i^V$, $W^O$ represent the weight coefficient obtained by network training. This module calculates the similarity between Q and K, multiplies V by the normalized weight, and realizes the feature enhancement of V. As shown in Figure 3, multiple Transformer based feature enhancement modules are used to realize the feature fusion pyramid.

Firstly, we use convolution to flatten the features of each layer into sequence information and take the middle layer Feat4 as the query of hierarchical feature fusion. Then, feature fusion is carried out with the upper and lower layers Feat3 and Feat5 to generate three groups of combinations with different feature information. At last, the results are sent to a TFE module to enhance self-attention, so as to obtain the final semantic features that integrate the underlying spatial information and deep semantic information. The above process can be expressed by the following formula.

$$
F3' = \text{TFE}(\text{Feat4}, \text{Feat3}, \text{Feat3}); F4' = \text{TFE}(\text{Feat4}, \text{Feat4}, \text{Feat4}); F5' = \text{TFE}(\text{Feat4}, \text{Feat5}, \text{Feat5})
$$
$$
F' = F3' + F4' + F5' \tag{2}
$$
$$
\text{F\_final} = \text{TFE}(F', F', F')
$$

Compared with the traditional FPN, the Transform can help overcome the limitations of the receptive field. Our proposed method can calculate the correlation between all positions of the input feature sequence, and enhance the target features by using the cross attention between different feature layers to establish the global interaction. In this fusion mode, by transmitting the semantic information to the multi-layer feature map, this paper not only fuses the deep and shallow features, but also adaptively realizes the interactive fusion of key feature information. Therefore, features obtained by TR-Siam are more significant, which can effectively enhance the robustness of the algorithm to deal with the interference and occlusion of similar targets.

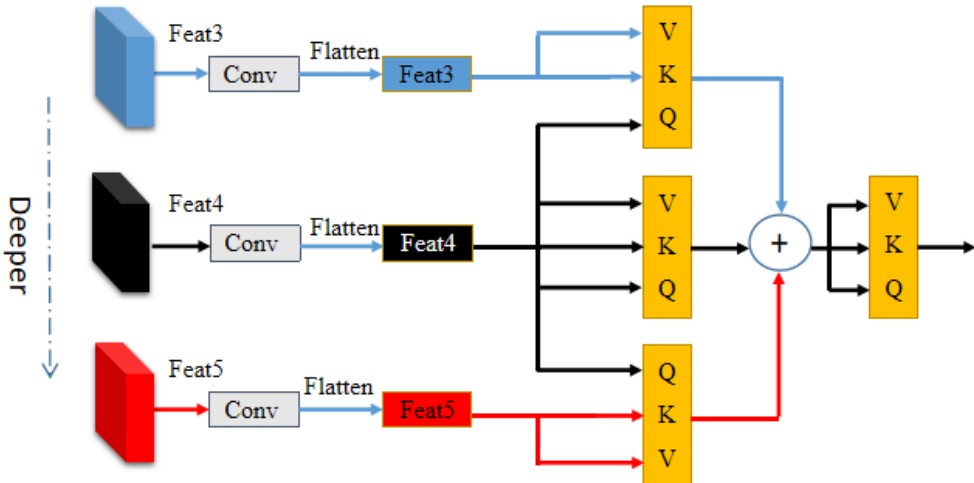

**Figure 3.** Transformer Feature Pyramid.Features form Feat3-Feat5 are flattened and fed into TFE blocks. TFN consists of 4 TFE blocks.Hierarchical information are extracted by 3 separate TFE blocks, and further enhanced by 1 additional TFE block.

### 3.3. Similarity Match Network

Cross correlation operation refers to the process of obtaining the feature response map by correlation operation with the search area and the template as the kernel. Inspired by reference [7], this paper introduces deep-wise cross correlation. Deep-wise cross correlation takes the search area as the input and the template as the kernel to obtain the response map channel by channel. It can greatly reduce the amount of parameters and ensure the tracking accuracy is not affected.

The tracking algorithm determines the positive and negative samples in the image and locates the target through the response map. By enhancing the rich feature information in the response map, the tracking accuracy can be improved. Therefore, we designed the response map enhancement module. As shown in Figure 4.

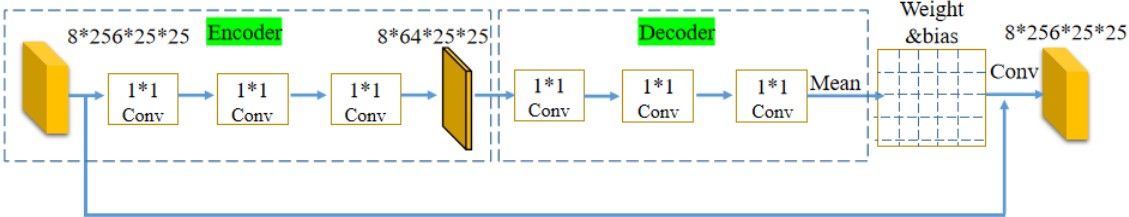

**Figure 4.** Response map Enhancement. The module consists of Encoder and Decoder. 8*256*25*25 in the figure represent batch*channel*height*width of the characteristic graph, respectively.

The proposed RmE method adjusts the response map by calculating its own mean and variance. After obtaining response map with the size of 8*256*25*25, we adjusted the response map through encoder and decoder. In the encoder, in order to reduce the amount

of computation, we used $1 \times 1$ convolution to adjust the response map and adjust the channel size from 256 to 64. In the decoder, we adjust the mean and variance of response maps in the same batch by batch normalization and restored the number of channels through $1 \times 1$ convolution. Because the targets of the same batch are same in the training process, the batch normalization of the response maps enables more target information to be obtained. The output is then averaged. The result was used as the weight of convolution kernel, and channel 0 of each data was used as the bias of the convolution kernel. Through this special convolution method, we adjusted the original response map to allow it to obtain global context information and enhance the discrimination ability of this network, so as to deal with similar object interference and significant appearance changes of the target.

## 4. Results

This section introduces the results of our method on UAV123, GOT-10K, LaSOT and OTB100 benchmarks [26–30]. To prove the effectiveness of the proposed blocks, we conducted ablation experiments and analyzed the effects of the TFN and RmE block. Additionally, we compared our method with baseline tracker SiamCAR and other advanced algorithms such as SiamRPN++, DaSiamRPN, SiamRPN and ATOM [31]. We visualized the heat maps and tracking results and analyzed the performance of TR-Siam in some challenging sequences such as Similar object and Occlusion.

### 4.1. Experimental Details

In the experiment, we used the pre-training parameters provided by SiamCAR to initialize TR-Siam. Training data includes COCO [32], GOT-10K, VID. In training process, the model is trained by SGD, the batch size is set to 32 and the sample size is 400,000 per epoch, 20 epochs in total. In the first five epoch, the warmup is used, for which the initial learning rate is 0.001, and is raised 0.001 per-epoch. The overall training time is 50 h. We tested TR-Siam on AMD3600X and NVIDIA RTX3060ti.

### 4.2. Algorithm Comparison

4.2.1. UAV123 Benchmark

UAV123 contains 123 image sequences, which were collected by a low altitude unmanned aerial vehicle. This benchmark had obvious characteristics, including tiny targets, similar objects, occlusion and so on. Success and Precision plots were used to evaluate the performance of the trackers. Precision plots refer to the percentage of the number of frames whose CLE is less than 20 pixels in the total frames. The formulas are as follows.

$$
\begin{aligned}
CLE &= \sqrt{\left(x_{pr} - x_{gt}\right)^2 + \left(y_{pr} - y_{gt}\right)^2} \\
f &= \begin{cases} 1, & CLE < 20 \\ 0, & CLE \geq 20 \end{cases} \\
Precision &= \frac{\sum_{i=1}^{N} f}{N}
\end{aligned}
\tag{3}
$$

$(x_{pr}, y_{pr})$ and $(x_{gt}, y_{gt})$ refers to the center position coordinates of the prediction box and ground truth. Success rate means the percentage of the number of frames whose overlap S is larger than 0.5 in the total frames. Overlap shows the IOU of the prediction box and ground truth of each frame. The result takes the average value of the whole frames. The formulas are as follows.

$$
\begin{aligned}
S &= IOU \frac{\left|R_{pr} \cap R_{gt}\right|}{\left|R_{pr} \cup R_{gt}\right|} \\
f &= \begin{cases} 1, & IOU \geq 0.5 \\ 0, & IOU < 0.5 \end{cases} \\
Success\ rate &= \frac{\sum_{i=1}^{N} f}{N}
\end{aligned}
\tag{4}
$$

The Success plots and Precision plots of the one-pass evaluation are shown in Figure 5. TR-Siam is observed to achieve a success rate of 0.635 and precision plot of 0.832. Addition-

ally, compared to HIFT, which also employs feature pyramid by Transformer, our proposed method achieved a substantial gain of 4.6% and 4.4% in terms of the success score and precision plots.

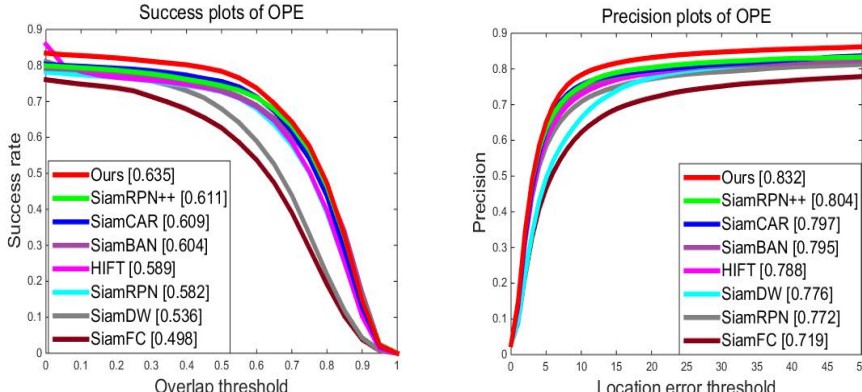

**Figure 5.** UAV123 comparison chart. The proposed algorithm TR-Siam performs favorably against state-of-the-art trackers. The left is Success plots, the right is Precision plots.

- Attribute-Based Evaluation

The performance of the five attributes is shown in Figure 6, demonstrating that our method obtained the best performance among the comparison methods. On the one hand, as for success rate, our method achieved better results, including Similar Object (0.579), Partial Occlusion (0.542), Full Occlusion (0.398), Aspect Ratio Change (0.595) and Viewpoint Change (0.651) On the other hand, as for the precision plot, we also attained better a performance in all challenges. Compared to the baseline tracker SiamCAR, our proposed method achieved significant performance gain in Similar Object and Partial Occlusion challenges with 5.4%, 3.9% in precision plot and 3.5%, 2.8% in success rate. In summary, the attribute-based evaluation proves our method can improve the performance when concerning similar object and occlusion.

In order to further demonstrate the improvement range of this algorithm, we visually display the accuracy of different algorithms under various attributes through radar charts. Figure 7 reports an attribute-based evaluation of state-of-the-art algorithms, illustrating that the TR-Siam performs much better than other competing trackers on all attributes. Especially, TR-Siam greatly improved the two key attributes of similar object interference and partial occlusion.

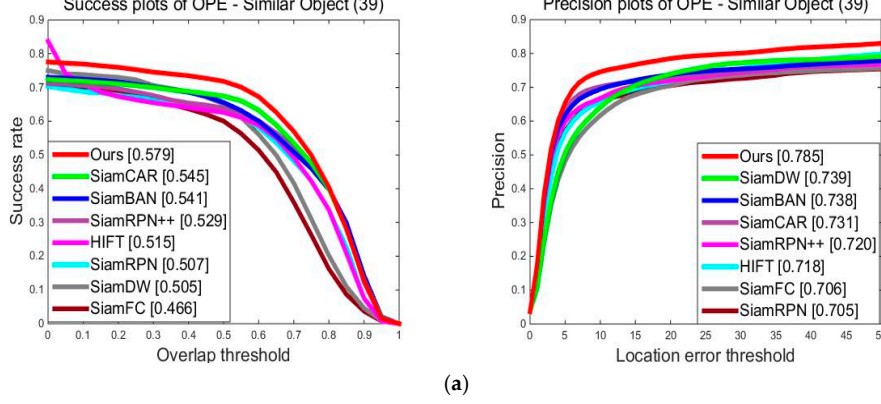

(**a**)

**Figure 6.** *Cont.*

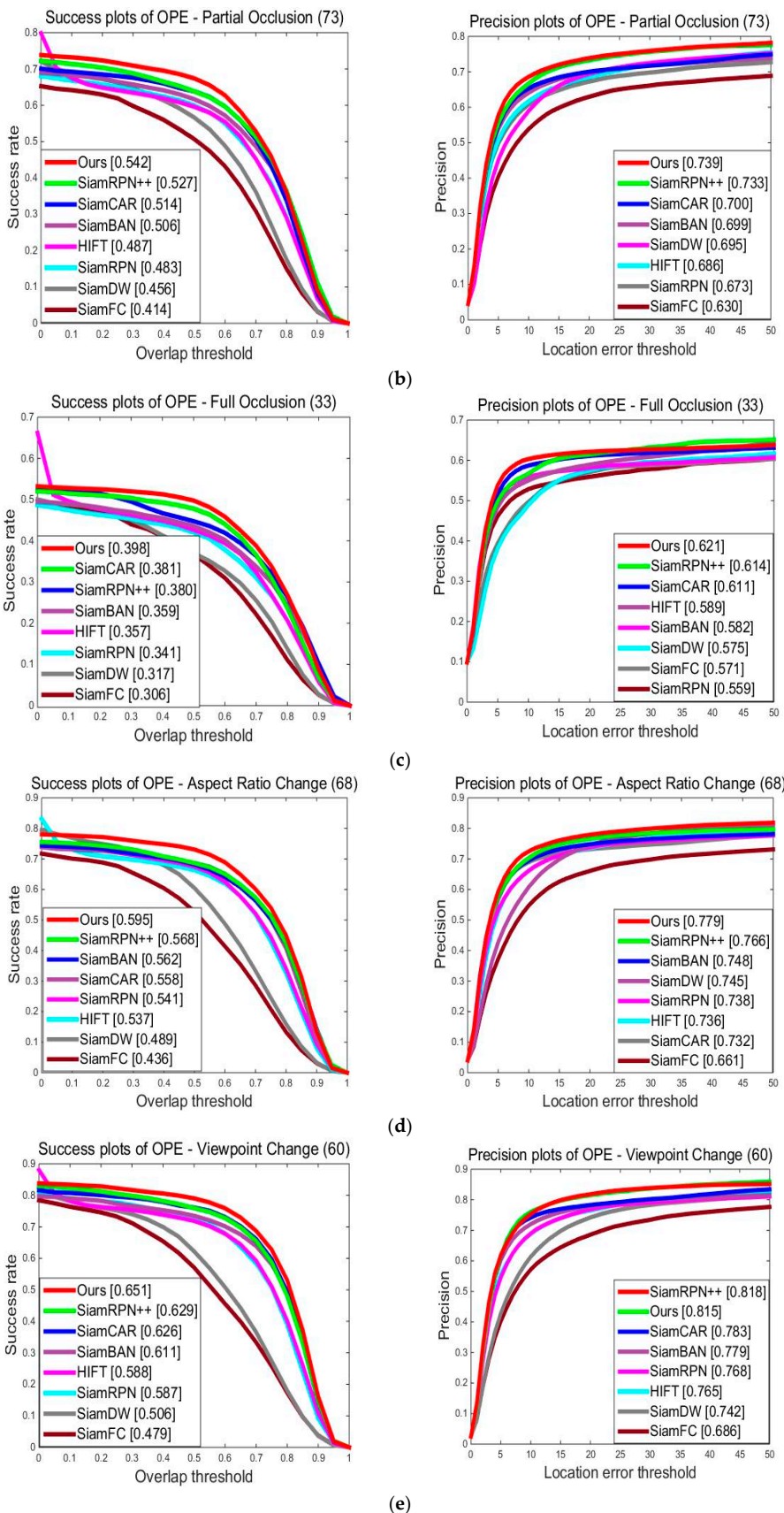

**Figure 6.** Experimental results corresponding to 5 typical scenarios, including (**a**) Similar Object; (**b**) Partial Occlusion; (**c**) Full Occlusion; (**d**) Aspect Ratio Change; (**e**) Viewpoint Change.

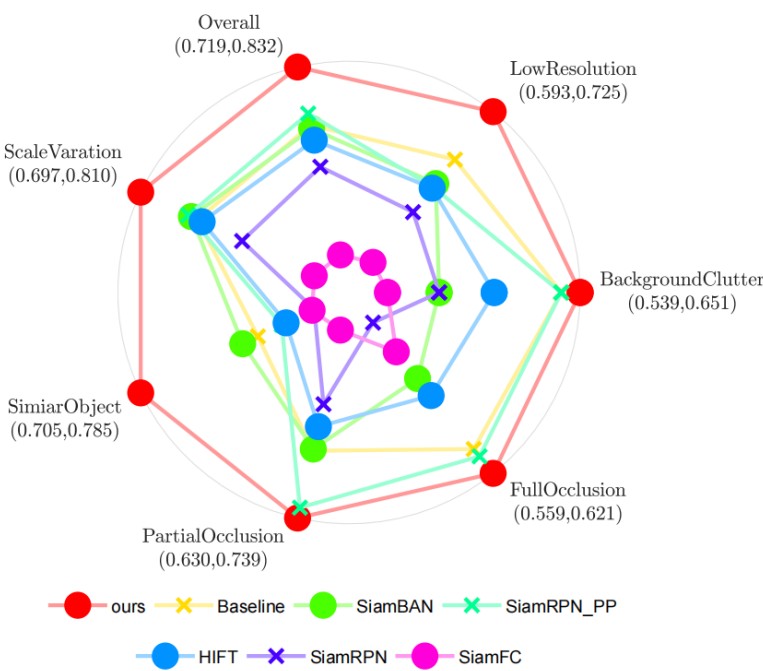

**Figure 7.** Precision scores of different attributes on the UAV123 data set. Take Overall as an example. In (0.719,0.832), 0.719 represents the algorithm with the lowest precision under this attribute, and 0.832 represents the algorithm with the highest precision under this attribute.

### 4.2.2. LaSOT Benchmark

The LaSOT data set is the most widely known benchmark for long-term tracking, consisting of 1550 sequences, which has more appearance changes, especially regarding long-term accumulation. This benchmark was selected to evaluate the robustness of trackers in long-term tracking. Success, Precision and Normalized precision are used to evaluate the performance of trackers. Normalized precision represents the average value of the overlap between prediction box and ground truth. Table 1 reveals that the proposed tracker achieves the highest target, demonstrating that the TFN and RmE module is an effective scheme for long-term tracking.

**Table 1.** Results on LaSOT benchmark. Red fonts indicate the top tracker ↑. Indicates that the larger the parameter value, the better the tracker performance.

| Tracker | Precision ↑ | Normalized Precision ↑ | Success ↑ |
|---|---|---|---|
| SINT [4] | 0.295 | 0.354 | 0.314 |
| ECO [14] | 0.310 | 0.338 | 0.324 |
| SiamFC [5] | 0.339 | 0.420 | 0.336 |
| MDNet [33] | 0.373 | 0.460 | 0.397 |
| SiamMask [34] | 0.469 | 0.552 | 0.467 |
| SiamRPN++ [7] | 0.491 | 0.569 | 0.495 |
| ATOM [31] | 0.497 | 0.605 | 0.499 |
| SiamCAR [18] | 0.507 | 0.600 | 0.507 |
| Ours | 0.527 | 0.606 | 0.528 |

### 4.2.3. GOT-10K Benchmark

GOT-10K is a challenging large-scale benchmark. To test the generalization of the tracking algorithm, there is no class overlap between the training subset and testing subset. GOT-10K employs Average Overlap (AO), Success Rate (SR 0.5, 0.75 mean threshold) and Frames Per Second (FPS) as the measurement. If FPS > 30, the algorithm meets real-time property. We evaluated TR-Siam on GOT-10K and compared it with other state-of-the-art trackers.

As demonstrated in Table 2, the proposed method achieved the best performance among all comparison algorithms. TR-Siam achieved a 2.7% higher performance on AO and 3.8% on $SR_{0.75}$ compared with the baseline tracker SiamCAR. Additionally, the FPS is 45, which meets the real-time property in 1080P resolution ratio.

**Table 2.** The evaluation on GOT-10K.TOP-2 results are highlighted in red and blue, respectively.

| Tracker | AO | $SR_{0.5}$ | $SR_{0.75}$ | FPS |
|---|---|---|---|---|
| KCF [12] | 0.203 | 0.177 | 0.065 | 94.66 |
| ECO [14] | 0.316 | 0.309 | 0.111 | 2.62 |
| SiamFC [5] | 0.374 | 0.404 | 0.144 | 25.81 |
| THOR [35] | 0.447 | 0.538 | 0.204 | 1.00 |
| SiamRPN [6] | 0.483 | 0.581 | 0.270 | 97.55 |
| SiamFC++ [35] | 0.493 | 0.577 | 0.323 | 160 |
| SiamRPN++ [7] | 0.518 | 0.618 | 0.325 | 35 |
| ATOM [31] | 0.556 | 0.634 | 0.402 | 30 |
| SiamCAR [18] | 0.569 | 0.670 | 0.415 | 50 |
| Ours | 0.582 | 0.683 | 0.457 | 45 |

*4.3. Visual Comparison of Tracking Results*

In order to intuitively show the tracking results, and illustrate the performance improvement in dealing similar object, occlusion and viewpoint change, we chose four image sequences from the UAV123 dataset and OTB100 dataset, which have the above challenges, compared with the baseline tracker SiamCAR and the other two state-of-the-art tracking algorithms.

Car1: The Car1 sequence is a video sequence with similar object interference and occlusion. First, similar object interference is shown. During 376–516 frames, disturbed by a car nearby, scale chaos appeared in the baseline tracker SiamCAR and the other algorithm. Particularly, in the 452 and 489 frame, the tracking box of DaSiamRPN and SiamRPN++ contained a similar object. Especially, after 516 frames, SiamRPN++ tracking appeared to drift.

Car1: Next, the role of occlusion is shown. SiamRPN++ drifted due to previous errors. The car began to be occluded at frame 749 until it was fully occluded at frame 781 and finally reappeared in frame 818. Only our proposed method could still locate and track the target effectively without being affected by occlusion.

Car7: In Car7 sequence, similar object interference occurs simultaneously in the process of occlusion. In frames 55–238, when the car is blocked and similar vehicles appear nearby, a scale error occurred in the baseline tracker SiamCAR and the other methods deviated from the target. After frame 316, when the target reappeared, the coincidence between TR-Siam and ground truth was still very high.

Box: As shown in Figure 8d, the main difficulty of the Box video sequence is that the Box is partially or completely occluded. In frames 302 to 315, when the target is partially occluded by the vernier caliper, all comparison algorithms can effectively track the target. However, as the difficulty increased, the box was placed behind the vernier caliper in frames 447 to 489 and was completely occluded. The comparison algorithm cannot track the target. Only our proposed algorithm can track the target robustly. It can be seen that the algorithm can still track the target robustly and accurately when the target is completely occluded.

Girl2: As shown in Figure 8e, in the Girl2 video sequence, the difficulty of the video sequence is that the girl is occluded by similar object. When the target begins to be occluded by similar targets in 107 frames, all algorithms look for the target. When the target is completely occluded in frame 111, tracking drift appears in SiamRPN++. From 111 frames to 129 frames, similar objects keep moving far away from the girl, and a tracking failure occurs in the comparison algorithm. However, the algorithm in this paper can still track the target correctly after the target reappears. This shows that our proposed algorithm

can effectively solve the problem of when the target is occluded by a similar object and ensure the robustness of the algorithm.

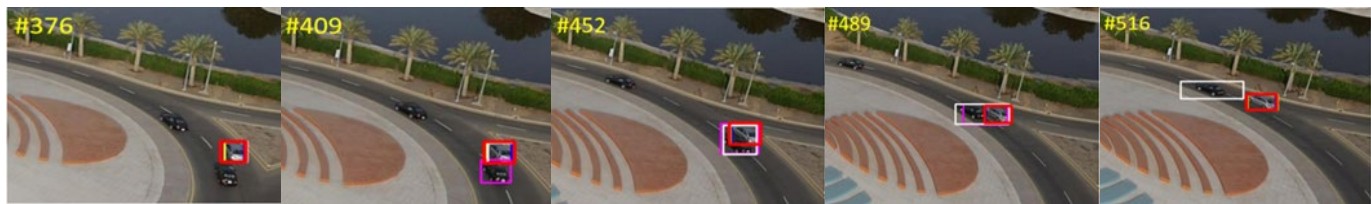

(**a**) Car1 subset

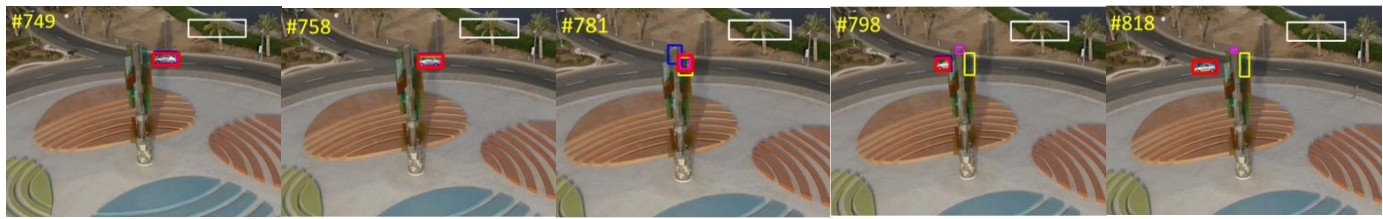

(**b**) Car1 subset

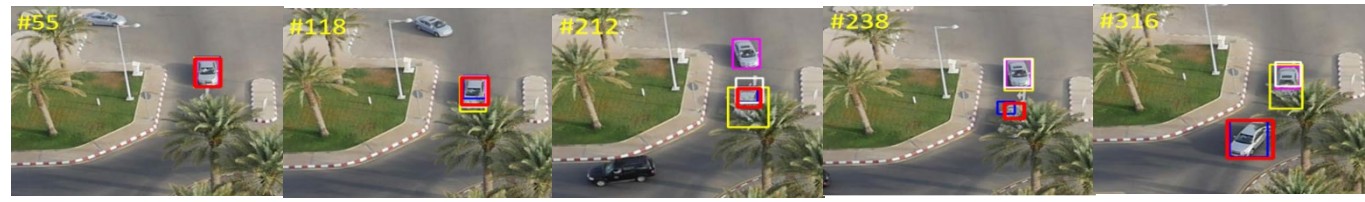

(**c**) Car7 subset

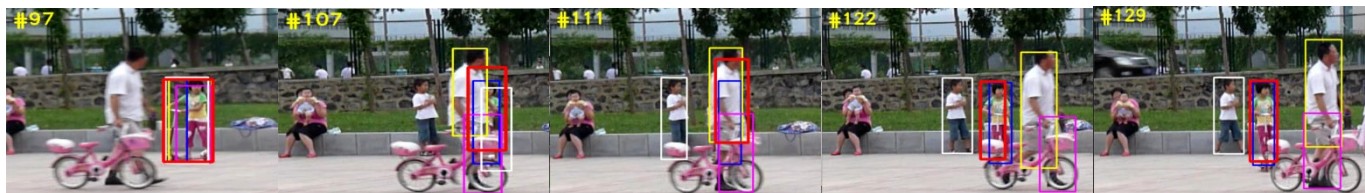

(**d**) Box subset

(**e**) Girl2 subset

**Figure 8.** Qualitative evaluation of the proposed TR-Siam and other state-of-the-art trackers on the UAV123 data set. From left to right and top to down are the tracking results on the videos of Car1, Car7, Box, Girl2. Red-Ours Yellow-SiamCAR White-SiamRPN++ Pink-DaSiamRPN Blue-Ground Truth.

### 4.4. Ablation Experiment

The TFN and RmE blocks are the core components of the proposed tracker. To further verify the effectiveness of TFN and RmE blocks, we conducted a number of ablation studies.

The purpose of this was to show the contribution of the module to the performance when tracker meets similar object interference and occlusion.

### 4.4.1. Heat Map Experiment

In order to show the difference between the baseline tracker and ours, and further illustrate the performance of the proposed module in dealing with similar object interference and occlusion, the interested objects of the algorithm are displayed in Figure 9.

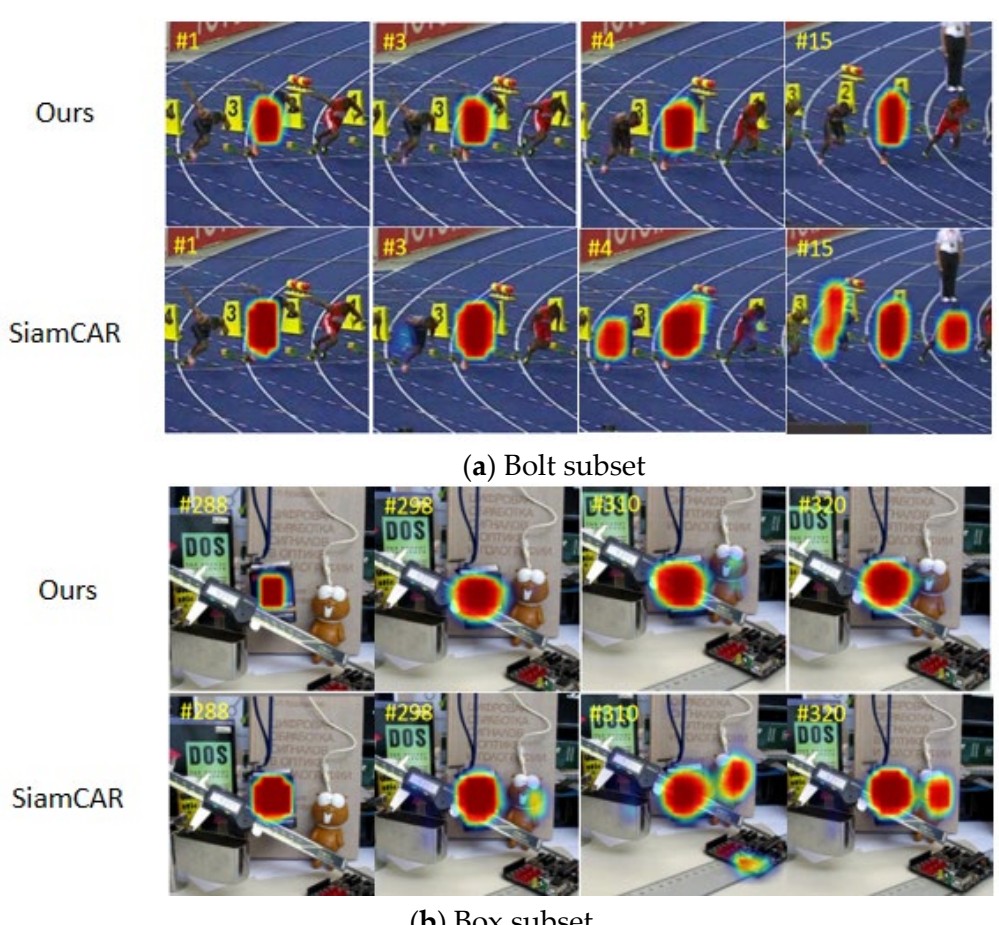

(**a**) Bolt subset

(**b**) Box subset

**Figure 9.** Heat map comparison between ours and SiamCAR.

The Bolt sequence is a video sequence with similar object interference and occlusion. By comparing heat maps, it can be found that our method can clearly distinguish between target and similar object, while baseline tracking SiamCAR is greatly affected by similar object interference focus on multiple similar objects. In the Box sequence, the attention of the baseline tracker SiamCAR begins to drift and it starts to pay attention to other objects in the area when the target is blocked. Our proposed method is not affected by this and only focuses on the target. This shows that our method will not experience attention drift when disturbed by a similar object and become blocked.

### 4.4.2. Data Experiment

To further verify the effectiveness of each module, we conducted an ablation study using three benchmarks. The aim of this was to show which module contributes to the performance of the tracker. As shown in Table 3, the symbol $\sqrt{}$ means a particular module has been used; $\times$ is the opposite.

Through the detailed ablation experiments in Table 3, it can be proved that TFN and RmE block can improve the tracking accuracy of the algorithm. Taking the UAV123 benchmark results as an example, there was a 0.8% improvement when using TFN alone,

a 1.3% improvement when using RmE alone and a 2.5% improvement when using both together. Meanwhile, when the template patch and the search patch were, respectively set to $127 \times 127 \times 3$ and $255 \times 255 \times 3$ resolutions, our method had little impact on operation efficiency because we compromised 5 FPS while improving the accuracy. Furthermore, our method uses AlexNet shallow backbone and can achieve the same accuracy as ResNet-50 deep backbone, while continuing to run at 130 FPS.

**Table 3.** A comparison of evaluation results of with or without TFN block and RmE block and different backbone on multiple datasets. Success represent accuracy on LaSOT; PRE represent Precision on UAV123 benchmark. AO represents accuracy on GOT-10K; FPS selects the operation results on GOT-10K data set.

| NO. | Backbone | TFN | RmE | Success | PRE | AO | FPS |
|-----|----------|-----|-----|---------|-----|-----|-----|
| 1 | AlexNet | √ | √ | 0.485 | 0.779 | 0.550 | 130 |
| 2 | ResNet-50 | × | × | 0.507 | 0.797 | 0.569 | 50 |
| 3 | ResNet-50 | √ | × | 0.507 | 0.806 | 0.572 | 47 |
| 4 | ResNet-50 | × | √ | 0.519 | 0.820 | 0.575 | 48 |
| 5 | ResNet-50 | √ | √ | 0.527 | 0.832 | 0.582 | 45 |

## 5. Conclusions

In the present article, a novel TR-Siam was proposed for solving the problem of tracking failure caused by similar object interference and occlusion. The TFN module can fusion deep and shallow features adaptively and efficiently, overcome the limitations regarding receptive field, achieve global context information fusion, and enhance the spatial and semantic information features. RmE block adjusts the response map, using statistical information to enhance the ability to distinguish between a target and background. A performance comparison of the four challenging benchmarks, including UAV123, GOT-10K, LaSOT and OTB100, demonstrated that the proposed approach can bring significant performance improvement. Compared to baseline tracker SiamCAR, the proposed method can effectively deal with the challenges of similar object and occlusion, and improve robustness and accuracy, without significantly affecting operation efficiency.

**Author Contributions:** Methodology, A.D.; Project administration, J.L.; Resources, X.W. and Y.Z.; Software, A.D.; Validation, Q.C.; Visualization, A.D.; Writing—original draft, A.D.; Writing—review & editing, Q.C. All authors have read and agreed to the published version of the manuscript.

**Funding:** This work was supported by the National Natural Science Foundation of China (No. 61905240).

**Informed Consent Statement:** Not applicable.

**Data Availability Statement:** Not applicable.

**Conflicts of Interest:** The authors declare no conflict of interest.

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
