# Peer review of "Visual Tracking with FPN Based on Transformer and Response Map Enhancement"

_applsci, doi:10.3390/app12136551_

Round 1

Reviewer 1 Report

The subject matter is interesting but does not bring anything new to the field of study. The authors presented the possibility of using the Siames Network in a specific business case. 

The paper presents more of an implementation case than a scientific paper. The scientific contribution is poor and does not present anything new beyond the information contained in the literature cited by the authors of the paper.

The authors used the literature correctly and quoted many current items. 

Despite the large number of results and summaries, the paper as it stands does not meet the requirements of a scientific publication.

Author Response

We appreciate your precious time and efforts on reviewing the manuscript. Please see the attachment.

Reviewer 2 Report

The new approach is novel and the results are clear.

The implementation performance as presented is comparable to a variety of other techniques.  The solution is effective and clear but its unclear what circumstances would merit the choice of this approach versus the others compared against.

Worthy of publication.

Author Response

We appreciate your precious time and efforts on reviewing the manuscript.

Thank you for your affirmation of our work.

Reviewer 3 Report

This manuscript stitches transformer networks to pairs of the deep convolutional neural network ResNet50 in a Siamese configuration. By enhancing the cross-correlation output, the authors were able to improve tracking accuracy in terms of identifying the target, and where the target might be with only minor compromises in tracking speed. Overall, this paper was well organized, straightforward and the results were consistent. There were few grammatical mistakes that can be easily corrected. My major complaint is it is hard to determine if <5% improvements were significant. It was also hard to determine if the network was really better than the others at high thresholds. The section on the similarity matched network needs to be rewritten for clarity, and claims of significance need to be substantiated with statistical tests.

Specific comments:

Line 180: “specially” should be specifically

Please increase font size in all figures.

Figure 3 and lines 203-204: Why Feat4 to query all layers? What about top down query? Feat 5 queries Feat 4, Feat 4 queries Feat 3 etc. Studies have shown that top down inference has benefits. Besides, Line 150 says that the last three outputs corresponding to Feat1, 2, and 3 were used. Why is there now Feat5?

Figure 2 and 3: Would help labeling in Figure 1 where these parts are or color coding them. It is mentioned in the text, but not blatantly obvious.

Line 211: “break through the limitation of receptive field”. Unclear. Does it mean to retain low level information like texture and color that would be lost higher in the hierarchy, or that attention in the transformer model would preserve contextual relationships across scales?

Similarity match network section: Line 226: what statistical information? Shannon information? Line 233: define B, C, H and W. Line 235: Don’t really understand what was done during the trimming and why? What is the kernel during this conv? I’m little confused here. Did you mean that the template output for Feat1 was cross-correlated with the test Feat1, template Feat2 with test Feat2 and so on? Then, some form of integration across all Feats and regularization in an autoencoder? I assume 256 stands for the 256 channels, but what were the 7x7 about? How did cross correlation between the template and test features give rise to 31x31xm? What’s m? I also couldn’t determine if the statistical characteristic was used to calculate information, or entropy. This section, together with Figures 4 and 1, requires more coherent explanation.

Line 266: what does CLE stand for? Don’t really understand the equation. If the (xpr, ypr)=(256, 256) and (xgt, ygt)=(128, 128), CLE = sqrt( (65536, 65536) + (16384, 16384) ) = (268.2, 286.2)? Shouldn’t it be the difference between (xpr, ypr) and (xgt, ygt) instead?

Figure 5: I assume that overlap threshold means the IOU number below which f=1? And the location error threshold means the maximum CLE that give f=1? Please clarify. If so, the present algorithm is only superior when the success and precision thresholds are relaxed, i.e., below 0.6 in the left plot and above 10 in the right plot. While 0.6 and 10 are beyond the cited thresholds of 0.5 and 20, it means that there is only a small margin (between 0.5 to 0.6, and between 10 to 20) of reliable data that outperforms others. In another words, it is meaningless to say that this network is better when the threshold is set to 0. By the same argument, on what basis is it meaningful to have better success rate when the threshold is say, set at 0.4?

Another point about Figure 5: The red trace is buried in thresholds beyond 0.6 and below 10 in the left and right plots. Please bring it to front so that performance can be evaluated when the criteria are tightened.

Lines 280: “substantial” by what standards? The term “substantial” has to be statistically substantiated, e.g. ANOVA. Or else some reference has to be cited as to why this small number is substantial. Otherwise, I contest the use of this word, and whether the data is significant at all. I don’t see the calculation that led to 4.6% and 4.4% either.

Figure 6: unclear how fully occluded objects were tracked. Full occlusion means invisible. Were the trajectories of invisible objects predicted and how? Or did the red box just disappear and reappear when the objects became visible again? Again, bring the red plots to front.

Lines 287-291. What do the numbers in parentheses mean?

Figure 7: Again, bring red to front. Blue should come next. All other colors go behind.

Line 325: show Frames 749-815.

Car1, Car4, Bike2, Truck2, Car7: Not obvious what these refer to. The images are too small to make out what’s what. Label them in Figure 6.

Line 414: change “pay” to “compromise”.

Conclusions: I would delete the word “significant” unless there is statistical evidence. Not clear as to what “adaptively” means in the phase “fuse deep and shallow features adaptively”. Again, “break through the limitation of receptive field” is unclear, and also please make clear what is meant by “statistical information”. Shannon information? Entropy? Bayesian inference? Joint probability?

Author Response

We appreciate your precious time and efforts on reviewing the manuscript.Thanks a lot for your comments and constructive suggestions. We seriously consider the comments from the reviewer, and make a concerted effort to improve the manuscript from several aspects. Please see the attachment.

Round 2

Reviewer 3 Report

Thank you for the revisions. I can see that effort was put into this.